# Decrease of Antimicrobial Resistance through Polyelectrolyte-Coated Nanoliposomes Loaded with β-Lactam Drug

**DOI:** 10.3390/ph12010001

**Published:** 2018-12-23

**Authors:** Lina M. Arévalo, Cristhian J. Yarce, José Oñate-Garzón, Constain H. Salamanca

**Affiliations:** 1Maestría en Formulación de Productos Químicos y Derivados, Facultad de Ciencias Naturales, Universidad Icesi, Calle 18 No. 122–135, Cali 760031, Colombia; linaarevaloh@hotmail.com (L.M.A.); cjyarce@icesi.edu.co (C.J.Y.); 2Grupo de investigación en Química y Biotecnología (QUIBIO), Facultad de Ciencias Básicas, Universidad Santiago de Cali, Cali 760031, Colombia; jose.onate00@usc.edu.co

**Keywords:** coated liposomes, layer-by-layer, ampicillin, antimicrobial resistance

## Abstract

Currently, one of the greatest health challenges worldwide is the resistance to antibiotic drugs, which has led to the pursuit of new alternatives for the recovery of biological activity, where the use of different types of nano-systems has shown an interesting potential. In this study, we evaluated the antibiotic activity of a model drug (ampicillin) encapsulated within coated-nanoliposomes on strains of *Staphylococcus aureus* with different antibiotic-resistance degrees. Hence, liposomes were elaborated by the ethanol injection method and were coated with a cationic polymer (Eudragit E-100) through the layer-by-layer process. Liposome characterization, such as size, polydispersity, zeta potential, and encapsulation efficiency were determined using dynamic light scattering and ultrafiltration/centrifugation techniques. Although biological activity was evaluated using three ATCC strains of *S. aureus* corresponding to ATCC 25923 (sensitive), ATCC 29213 (resistant) and ATCC 43300 (very resistant). The results showed changes in size (from ~150 to 220 nm), polydispersity (from 0.20 to 0.45) and zeta potential (from −37 to +45 mV) for the coating process. In contrast, encapsulation efficiency of approximately 70% and an increase in antibiotic activity of 4 and 18 times more on those *S. aureus*-resistant strains have been observed.

## 1. Introduction

Currently, antimicrobial resistance has been considered one of the greatest challenges in medicine according to the World Health Organization because this problem leads to the conventional therapy for many infectious diseases becoming difficult to treat [1]. According to the aforementioned paragraph, several types of research focused on (i) characterization of different molecular mechanisms involved in the generation of the drug resistance [2]; (ii) search for new molecules with antimicrobial potential [3]; and (iii) use of particle nanotechnology as a new tool to improve antibiotic performance, both new drugs and those that are practically obsolete [4,5]. Although there is a great diversity of microorganisms associated with this problem, *Staphylococcus aureus* has been highlighted to be one of the most relevant due to its high recurrence, making it one of the primary pathogens with resistance problems worldwide [6]. Such microorganism has generated several specific mechanisms of resistance, where the production of specialized enzymes (β-lactamases) is one of the most used complexes because it has inactivated many conventional β-lactam antibiotic drugs [7]. In contrast, new tools of nanotechnology have allowed the development of new systems for the delivery of traditional antibiotics with drug-resistance issues, where nano-emulsions [8,9,10], polymer nanoparticles [11], lipid nanoparticles [12,13], dendrimers [14,15], and liposomes have been highlighted as the most used systems [4,16,17,18]. Liposomes have shown very interesting results due to their biocompatibility features, ease of processing, and versatility in modifying and conferring new properties [19,20]. In this regard, liposomes coated with polymers have been called in different ways, such as ‘colloidosomes’ or ‘stealth liposomes’ when ionic polymers or polymer derivative of polyethylene-glycol is used, respectively, which could be an interesting alternative in overcoming drug-resistance problems [1,21,22,23]. Moreover, this strategy of liposomes coated with polymers is useful in projecting pharmaceutical preparations for reconstitution, oral, and topical dosage forms. With the aim of contributing to finding solutions to this complex problem, we have been working in our laboratory regarding the use of nanoparticle technology as a potential strategy to help recover the biological activity of antibiotic drugs with resistance problems. Hence, several types of polymer–drug nanocomplexes have been evaluated on *S. aureus* strains with different resistance degrees [11,24]. However, in this work, we evaluated another alternative of nanoparticles corresponding to nano-liposomes that were modified superficially with a cationic polymer derived from Eudragit E-100.

## 2. Materials and Methods

### 2.1. Materials

Ampicillin (Fersinsa Gb) was supplied by Tecnoquímicas S.A. (Cali, Colombia), and was used as received. Soy lecithin or phosphatidylcholine Epikuron 200™, Mw = 786 g/mol from Cargill (Wayzata, MN, USA), dioleoyl-phosphatidyl-ethanolamine (DOPE, Mw = 744.03 g/mol) and cholesterol (Mw = 386 g/mol) were purchased from Avanti Polar Lipids (Alabaster, AL, USA). Eudragit^®^ E-100 from Evonik (Darmstadt, Germany) was used for liposomes surface coating. Ethanol USP grade was purchased from Sigma-Aldrich (St. Louis, MO, USA). (Merck) Ultrapure water was supplied from an Elix Essential Millipore^®^ purification system, with a mean conductivity value of ~1 μS/cm.

### 2.2. Preparation of Liposome Systems

#### 2.2.1. Experimental Design Optimization

A complete factorial design was performed to establish whether some process variables commonly used in the ethanol injection method [25], corresponding to (i) ionic strength, (ii) aging time, and (iii) membrane pore size, could significantly affect the physiochemical features of liposomes (dependent variables), such as particle size, polydispersity, and zeta potential. Statistical analysis was performed using the Minitab 17 software. The complete number of runs (treatments) that composed the experimental design is summarized in Table 1.

The liposomes were prepared on the basis of a sequential process defined in several steps. Step 1 (preparation of organic phase): Ethanolic solutions of lecithin, Epikuron 200™ (1.3 mg/mL), cholesterol (0.64 mg/mL) and DOPE (1.23 mg/mL) were elaborated, from which volumes of 42.3, 42.4, and 15.3 μL were taken, respectively, to obtain 100 μL of the lipid mixture. Step 2 (phase mixture): 100 μL of organic phase was slowly added to 100 μL of different aqueous media (Ultra-pure water, PBS pH 7.4, 1 mM and PBS pH 7.4, 10 mM), which were stirred (in vortex) for 1 min and left ‘aging’ at different times (5 and 20 min). Step 3 (formation of liposomes): The resulting mixture between the organic phase and aqueous media was diluted in 300 μL of the respective aqueous media. Step 4 (liposome purification): The diluted mixture was centrifuged (9000 g or 10000 rpm) in a micro-centrifuge (Hettich RCF 10538) for 6 min, using ultrafiltration tubes (Eppendorf) with different pore sizes (MWCO 10 kDa and MWCO 30 kDa). Subsequently, the fractions of purified liposomes were extracted and resuspended in 500 μL of the respective aqueous media. Each assay was performed in triplicate.

#### 2.2.2. Preparation of Liposomes Loaded with ampicillin

In this case, the liposomes were prepared in a similar way as described in the previous section. However, some variations were made. In step 2, the addition of the organic phase (lipid mixture) was added to 100 μL of ampicillin solution with a concentration of 6 mg/mL, using PBS (pH 7.4, 10 mM) as media. The aging time was only 5 min, and the pore size membrane used in step 4 was MWCO 30 kDa.

#### 2.2.3. Liposome Surface Modification

First, the aqueous solutions of Eudragit^®^ E-100 were prepared at different concentrations of 0.3%, 0.5%, and 0.7% (% *w*/*v*), fitting the media pH to 4.0 with 0.1 M HCl. Subsequently, 1 mL of the respective polymeric solutions of Eudragit E-100 was added on 1 mL of liposomal dispersion loaded with ampicillin (previously prepared) at a rate of 50 μL/min. Subsequently, the mixture was left under constant magnetic stirring at 300 rpm for 8 h in a closed vessel. Finally, it was centrifuged at 10,000 rpm for 2 min, using ultrafiltration tubes (Eppendorf) with 30 kDa cut-off.

### 2.3. Characterization of Liposome Systems

#### 2.3.1. Zeta Potential and Size Measurements

Particle size and zeta potential were determined using a Zetasizer nano ZSP (Malvern Instrument UK) with a red laser (633 nm) He/Ne. The particle size was measured using dynamic light scattering with an angle scattering of 173° at 25 °C, using a quartz flow cell (ZEN0023), whereas the zeta potential was measured using a disposable folded capillary cell (DTS1070). The instrument reports the particle size as the mean particle diameter (z-average) and PDI ranging from 0 (monodisperse) to 1 (very broad distribution). All measurements were performed by triplicate, after an appropriate dilution (5:5000, *v*/*v*) of the liposome suspension in ultra-pure water and were reported as the mean and standard deviation of measurements made from freshly prepared liposomal dispersions.

#### 2.3.2. Encapsulation Efficiency

In this case, the amount of ampicillin that was not retained in step 4 of the liposome preparation process (non-encapsulated ampicillin), which was contrasted against a calibration curve (*R*^2^ = 0.9995), was previously determined using UV spectroscopy at 256 nm and 25 °C (Shimadzu, Kyoto, Japan). Therefore, the encapsulation efficiency was calculated, on the basis of Equation (1).
(1)$$EE=\frac{{\left[Drug\right]}_{encapsulated}}{{\left[Drug\right]}_{non-encapsulated}+{\left[Drug\right]}_{encapsulated}}\times 100,$$
where the [*Drug*]*_total_* corresponds to the total amount of ampicillin before ultrafiltration step, whereas [*Drug*]*_encapsulated_* = [*Drug*]*_total_* − [*Drug*]*_non-encapsulated_*.

### 2.4. Stability of Liposomes

The stability of the coated and non-coated liposomes was performed using a stability chamber at 40 ± 1 °C, where the change in liposomal size was evaluated for 7 days in triplicate.

### 2.5. Antimicrobial Susceptibility Test

Microbial susceptibility tests were performed based on the Clinical and Laboratory Standards Institute: CLSI Guidelines [26]. Bacteria were inoculated in Mueller Hinton broth and incubated overnight at 37 °C. Subsequently, the culture was diluted in Mueller Hinton broth until an OD_625_ of 0.1 was reached (approximately 1 × 10^8^ CFU/mL). The culture was diluted again by a factor 1:100. Subsequently, 50 μL of culture was incubated for 18–20 h into 96-well plates at 37 °C with 50 μL of antibiotic to reach a final inoculum of approximately 5 × 10^5^ CFU/mL. Ampicillin (Amp), ampicillin loaded in non-coated liposomes (NCL) and ampicillin loaded in coated liposomes (CL), were applied at 12 different serial concentrations from 0.09 to 201.7 μg/mL. After incubation, the minimum inhibitory concentration (MIC) was determined by visual analysis.

## 3. Results and Discussion

### 3.1. Optimization of Liposome Preparation Process

The process variables (ionic strength of the aqueous media, membrane pore size, and aging time) have been found to significantly affect the response variables (size, polydispersity index [PDI], and zeta potential) and to interact with each other. Generally, the adjusted models presented a good correlation coefficient, which allows us to establish with confidence the best conditions for liposome elaboration. The results of the statistical analysis of the factorial model are shown in Figure 1.

Figure 1A shows that the aging time is the condition of the elaboration process that most affects the liposome size, where a change of ~20 nm was observed. Other factors, such as membrane pore size and ionic strength of the aqueous media, seem not to significantly affect the liposomal size. This result is consistent because it has been found that liposomes of low phospholipid concentration (nM order), the process of vesicular stabilization depend on two relaxation periods of time. The lipid monomers moving easily from aggregates are formed in a short period, and the curvature of the lipid bilayer of the liposome is generated and stabilized over a slower period [27]. Therefore, it could be expected that agglomeration of the phospholipids will be greater at the interface for longer aging times; thus, vesicles will form with a larger size. 

In contrast, the PDI always has low values (<0.3) and was not affected significantly by either the aging time or ionic strength (Figure 1B). In the case of zeta potential, this parameter was found to be considerably affected by the process variables used in the liposome elaboration (Figure 1C). With regard to the ionic strength of the aqueous media, a change in the zeta potential value was observed from ~−47 to ~−35 mV, which could be explained by the compression effect of the electrical double layer, given by the loss of the diffuse layer in the system [28,29]. Conversely, the decrease in the membrane pore size leads to a change in the zeta potential of ~−48 to ~−36 mV. This change could be explained by rheological variations generated during the centrifugation/filtration process, where smaller pore causes greater shearing effects that may affect the electrical double layer through ion desorption from the liposomal surface. Finally, process conditions corresponding to ionic strength of 10 mM for the PBS buffer pH 7.4, membrane pore size of 30 kDa and aging time of 5 min were selected because of such conditions, and the liposomes were obtained with adequate size (~140 nm), low polydispersity (<0.3), and a high negative zeta potential value, which is necessary for the layer-by-layer coating process using the cationic polymer (Eudragit E-100).

### 3.2. Liposome Surface Modification

First, the change in particle size and zeta potential of the non-coated liposomes (without ampicillin) were determined against the pH (Figure 2A). Likewise, the effect of Eudragit E-100 polymeric concentration for zeta potential and viscosity was evaluated (Figure 2B) to establish the best conditions before loading the drug and modifying the liposomal surface. Secondly, the change in particle size, PDI, and zeta potential for liposomal systems loaded with ampicillin before and after the coating process was determined. The results are shown in Figure 3.

Figure 2A shows that the non-coated liposomes in aqueous solution change both the particle size and the zeta potential with respect to the media pH. With regard to the liposomal size, the particle size tends to remain constant at ~125 nm at pH values between 4.0 and 5.5, whereas an apparent transition occurs at pH > 5.5, wherein the liposomal size increases to ~160 nm until pH 7.4. The zeta potential increased with the increase in the media pH, shifting from ~−18 mV (pH 4.0) to ~−40 mV (pH 7.4). These results are consistent, considering that at pH 7.4, the phosphatidylcholine has a fraction of carboxylate groups, which begin to neutralize with the decrease in pH, affecting the electrical double layer in the liposomal surface [30,31]. Therefore, the loss of surface charge leads to a decrease in the electrostatic repulsion in lecithin heads, forming a more compact and smaller surface. In contrast, Figure 2B shows that the zeta potential of Eudragit E-100 polymer in acidulated aqueous solution (pH ~ 4–5) is positive and indifferent of polymeric concentration (~+50 mV). This result is consistent because this polymer derived from amino-alkyl methacrylate can be ionized in the acid medium, becoming a polyelectrolyte positively charged interface [32]. Conversely, the effect on the viscosity of the Eudragit E-100 polymeric solutions showed a marked increase above a concentration of 1.2% *w*/*v*. Based on these results, the best conditions of polymeric concentration to perform the liposomal coating are between 0.1% and 0.7% *w*/*v*. Therefore, the concentrations corresponding to 0.3, 0.5 and 0.7% *w*/*v* were selected for the liposome surface modification stage. 

Figure 3A shows the change in the zeta potential of the liposomes concerning the coating process from ~−40 mV (non-coated liposomes) to ~+50 mV (coated liposomes). Such change is an indicator that surface modification occurred through the polymer deposition process layer-by-layer [33]. The change in the size of ~150 nm (non-coated liposomes) to a size greater than ~200 nm (coated liposomes) also suggested the occurrence of such surface modification (Figure 3B), where the increase in polymer concentration Eudragit E-100 led to a slight increase in liposomal size. With regard to PDI (Figure 3C), a change from 0.2 (non-coated liposomes) to values between 0.4 and 0.5 (coated liposomes) was observed, suggesting a modification in the size populations, which passed from a monodisperse (~150 nm) to a polydisperse system (~200–250 nm) with slightly different sizes. This result could be explained because of the random deposition of the cationic polymer on the liposomal surface, where different amounts of polymer chains are adhered with different types of conformations, obtaining coated liposomes with extended or coiled polymer chains, as shown in Figure 3E. Finally, the liposomal coating process did not significantly affect the encapsulation efficiency of ampicillin, which remained approximately 70% in both uncoated and coated liposomes (Figure 3D).

### 3.3. Stability of Liposome

The results of the stability study showed that the liposomes coated with Eudragit E-100 polymer were more stable than the non-coated liposomes (Figure 4). This result could be explained considering the electrostatic stabilization effect [29], which increases with the adsorption of the polymer on the liposome surface. With regard to the amount of polymer used to coat the liposome, concentrations of 0.3% *w*/*v* and 0.5% *w*/*v* showed higher stability, whereas concentration of 0.7% *w*/*v* showed a marked increase in the liposomal size due to a competitive effect generated between polymeric adsorption on the liposome surface and aggregation between the same polymer chains. For this reason, liposomes coated with Eudragit E-100 at 0.5% were chosen for biological evaluation.

### 3.4. Antimicrobial Susceptibility Test

Figure 5 shows the minimum inhibitory concentrations (MIC) of ampicillin (Amp), ampicillin and Eudragit E-100 (coat polymer) (Amp-Eudragit E-100), ampicillin loaded in non-coated liposomes (Amp-NCL) and ampicillin loaded in coated liposomes (Amp-CL) on strains of *S. aureus* with different degrees of antimicrobial resistance. In addition, the MIC of the coating polymer (Eudragit-E-100) and the empty liposomes as negative test controls are included. In the case of non-encapsulated ampicillin, different MIC values were found corresponding to 0.19, 6.3, and 25.2 μg/mL for strains of *S. aureus* ATCC25923, ATCC29213, and ATCC43300, respectively. These values were consistent considering that the strain of *S. aureus* ATCC25923 is sensitive, whereas strains ATCC29213 and ATCC43300 have different degrees of antimicrobial resistance based on the Clinical and Laboratory Standards Institute: CLSI Guidelines [26].

In the case of the ATCC25923-sensitive strain, ampicillin could inhibit the synthesis of peptidoglycans in the bacterial cell wall due to the specific interaction with the penicillin-binding protein (PBP, specifically PBP1 and PBP3) affecting bacterial growth [34].

In contrast, the ATCC29213 strain showed a resistance effect against ampicillin because such strain is producing β-lactamase enzymes. Although the ATCC43300 strain showed the highest degree of resistance against ampicillin because of the production of β-lactamase enzymes, it also has a specific gene (mecA), which encodes an analogous PBP2a protein that has a lower affinity with the β-lactam drug, affecting its mechanism of pharmacological action [35,36].

Concerning Amp-NCL, the antimicrobial activity slightly increased in the three strains of *S. aureus*. This result may be explained considering that the liposomal systems can transport the antibiotic to the microorganism, preventing the degradation caused by β-lactamase enzymes, where the liposome adheres to the microorganism surface and is subsequently incorporated into the microorganism by an endocytosis mechanism [37].

In contrast, the ampicillin loaded in the liposomes coated with Eudragit E-100 (Amp-CL) displayed a marked increase in the antimicrobial activity, where the MIC was reduced from 0.19 to 0.09 μg/mL in the ATCC25923 strain, from 6.3 to 1.53 μg/mL in the ATCC29213 strain, and from 25.2 to 1.41 μg/mL in the ATCC43300 strain. The last result is very interesting because MIC < 2 μg/mL are classified as a penicillin-sensitive strain [26]. Therefore, the increase of 18 times in the antibacterial activity on *S. aureus* ATCC43300 strain is a significant and promising result because it shows that the use of nanotechnological systems can help recover drug activity with problems of antimicrobial resistance. 

It is important to note that both the liposomes alone and the polymeric material Eudragit E-100 did not show marked antimicrobial activity, just at the maximum evaluated values (512 μg/mL). This result suggests that the materials used as ampicillin vehicles are innocuous. All these results show that nanoparticulate systems are in fact an alternative promissory in the treatment of infectious diseases with resistance problems because such nano-systems allow the recovery of antimicrobial activity with simple formulating strategies.

## 4. Conclusions

The preparation method used allowed us to obtain liposomes with nanometric sizes of approximately 150 nm, which began to aggregate from the first day of preparation. In contrast, the polymeric coating with Eudragit E-100 led to an increase of approximately 50 nm in size and to an inversion of the liposomal zeta potential, passing from a negative to a positive surface. Likewise, such surface modification in the liposomes showed an improvement in the stability, being better at polymer coating concentrations of 0.3% *w*/*v* and 0.5% *w*/*v*. The polymeric coating did not also affect the encapsulation efficiency of ampicillin, which was approximately 70%. With regard to the antibacterial effect, a marked increase was seen with the liposomal coating, particularly in resistant *S. aureus* strains, which is a very important and promising result that shows that the use of nanoparticle systems can be an interesting alternative to combat the current problems of antimicrobial resistance.

## Figures and Tables

**Figure 1 pharmaceuticals-12-00001-f001:**
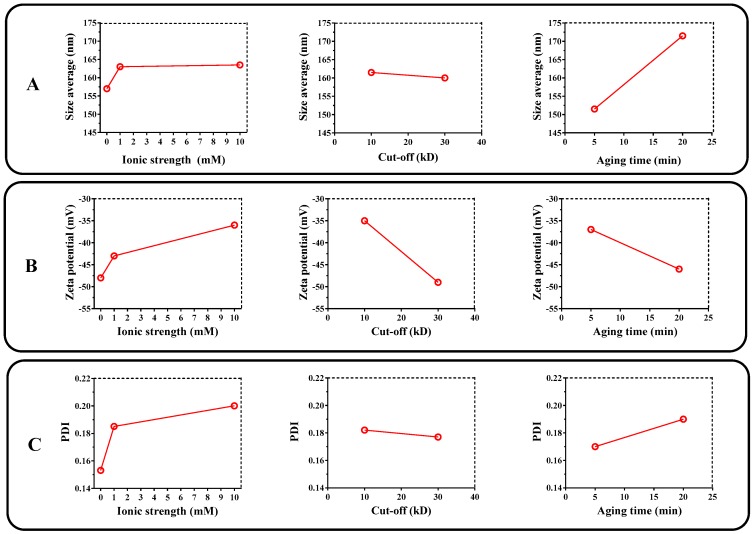
Effects plot of the process variables of liposome elaboration (ionic strength, membrane cut-off, and aging time) regarding the physicochemical features of liposomes. ((**A**) particle size, (**B**) zeta potential, (**C**) polydispersity index [PDI]).

**Figure 2 pharmaceuticals-12-00001-f002:**
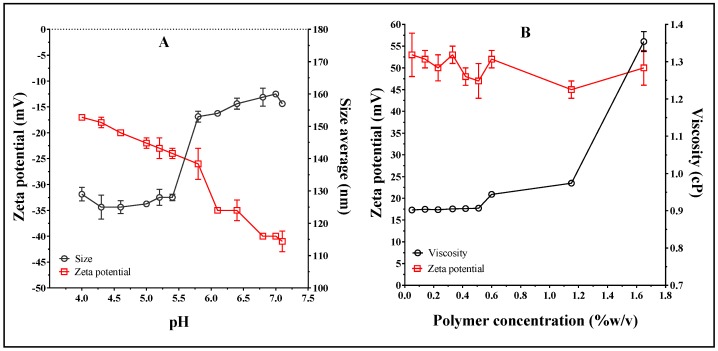
(**A**) Effect of media pH on the size and zeta potential of non-coated liposomes. (**B**) Effect of coating polymer (Eudragit E-100) concentration on zeta potential and viscosity in aqueous media.

**Figure 3 pharmaceuticals-12-00001-f003:**
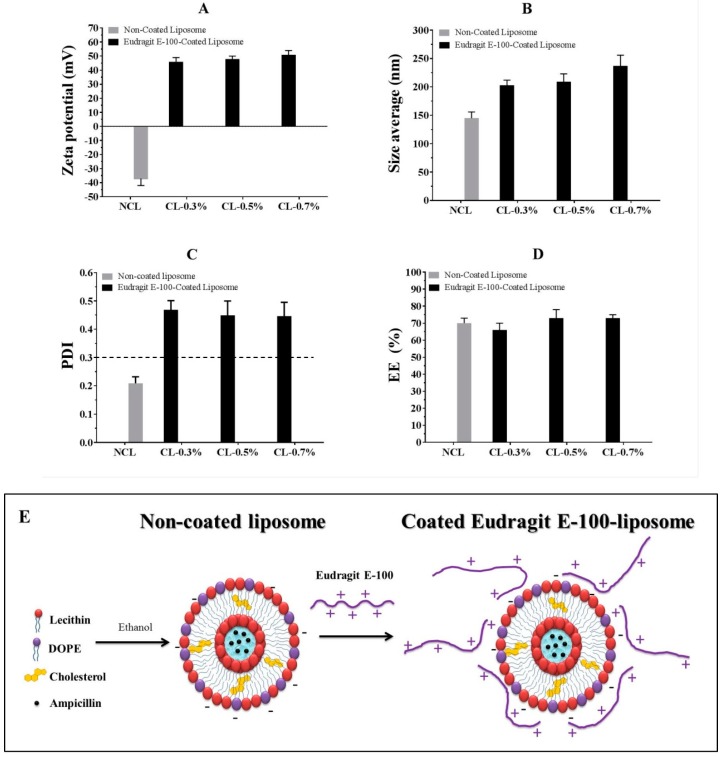
Change in (**A**) zeta potential, (**B**) particle size, (**C**) polydispersity index (PDI) and (**D**) encapsulation efficiency (EE) for liposomal systems loaded with ampicillin before and after the coating process. (**E**) Representative scheme of the liposomal coating process. NCL, non-coated liposome; CL, coated liposome.

**Figure 4 pharmaceuticals-12-00001-f004:**
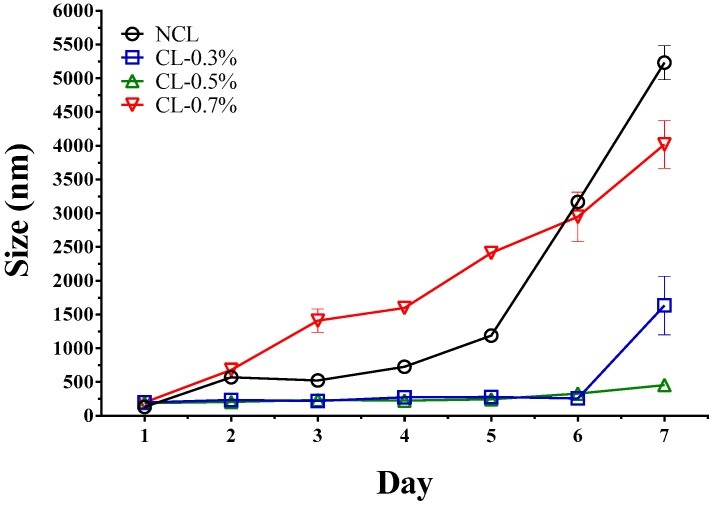
Evaluation of liposomal size change concerning time.

**Figure 5 pharmaceuticals-12-00001-f005:**
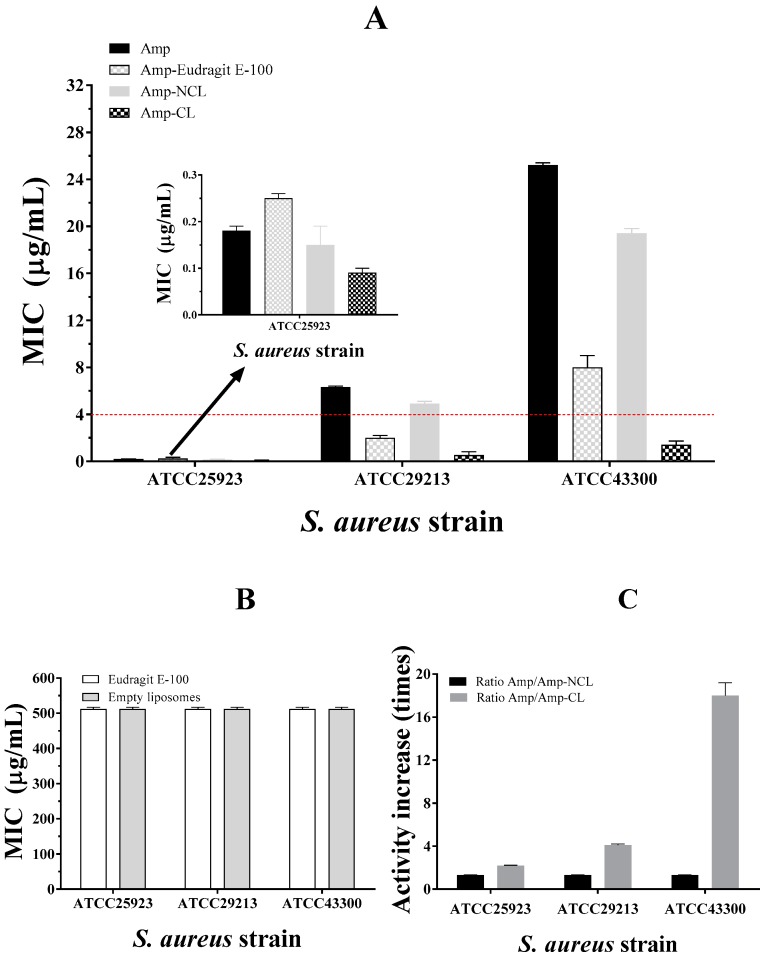
Minimum inhibitory concentration (μg/mL) of (**A**) ampicillin (Amp), ampicillin and Eudragit E-100 mixture (Amp-Eudragit E-100), ampicillin loaded in non-coated liposomes (NCL) and ampicillin loaded in coated-liposomes (CL) and (**B**) empty liposomes and Eudragit E-100, against three strains of *S. aureus* with different antimicrobial resistance degrees. (**C**) Comparison of the increase in the antimicrobial activity of ampicillin loaded between non-coated and coated liposomes.

**Table 1 pharmaceuticals-12-00001-t001:** Summary of the experimental design used.

Run	Ionic Strength (mM)	Cut-Off (MWCO)	Aging Time (min)	Run	Ionic Strength (mM)	Cut-Off (MWCO)	Aging Time (min)
1	1	30 kDa	5	19	10	10 kDa	20
2	0	30 kDa	20	20	10	30 kDa	5
3	1	10 kDa	5	21	10	30 kDa	5
4	1	30 kDa	20	22	0	10 kDa	20
5	10	10 kDa	5	23	10	30 kDa	20
6	1	10 kDa	20	24	10	10 kDa	20
7	10	10 kDa	5	25	1	10 kDa	5
8	10	10 kDa	5	26	1	30 kDa	20
9	0	30 kDa	5	27	0	10 kDa	5
10	1	30 kDa	20	28	1	10 kDa	20
11	10	10 kDa	20	29	1	10 kDa	20
12	0	30 kDa	5	30	0	10 kDa	5
13	10	30 kDa	20	31	0	30 kDa	5
14	1	30 kDa	5	32	10	30 kDa	20
15	10	30 kDa	5	33	1	30 kDa	5
16	1	10 kDa	5	34	0	10 kDa	20
17	0	10 kDa	20	35	0	30 kDa	20
18	0	10 kDa	5	36	0	30 kDa	20

## Data Availability

The datasets generated and analyzed during the current study are available from the corresponding author on reasonable request.

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
