# Peer review of "Decrease of Antimicrobial Resistance through Polyelectrolyte-Coated Nanoliposomes Loaded with β-Lactam Drug"

_pharmaceuticals, 2018, doi:10.3390/ph12010001_

Round 1
Reviewer 1 Report
This manuscript reports on ehnanced efficiency of ampicillin when this antibacterial drug is encapsulated into DOPE nanocarriers. The experimental design is carried out with accuracy and the resulrs obtained agais cultures of S. aureus are very promising.
However:
- the authors do not explain the choice of this phospholipid and indeed it is rather unusual to use DOPE alone to obtain liposomes, since it is well documented that its packing parameter is not suitable for the formation of stable bilayers (i.e. aggregates with zero local curvature). For this reason DOPE is usually employed in conjuction with DOPC or other bilayer forming lipids. Nevertheless, DOPE is also very sensitive to pH and it is possible that some of the systems prepared in this work actually posses the structure of closed bilayers. In my opinion the physico-chemical characterization carried out by the authors of this manuscript is nor extended enough to talk about liposomes in all the experimental conditions chosen for preparation. Therefore I would rather call the carriers nano-aggregates or nano-formulations, since the liposome structure is not ascertained with sufficient evidence.
- Another week point of this work is that the authors do not discuss in depth the implications of using a positively charged polymer (Eudragit E-100) for the coating. As all cationic molecules, this polymer is likely to have antibacterial effect by itself, but it is not possible to see the extent of the antimicrobial activity exerted by Eudragit E-100 alone, since the corresponding reference tests on S. aureus are not provided.
In summary, this work is likely to be of interest for researcher working on pharmaceutical preparations and corresponding antibacterial trials, but the above mentioned questions have to be addressed before publication.
Minor points
- though the subject of antibiotic resistance is a major concern in modern pharmacetics, the idea of encapsulationg antibiotics into lipid based nanocarriers is not new and the literature cited by the autors in this respect is rather incmplete.
- Figure 1 could be more conveniently relaced by a table, since it doesn't make much sense to show graphs with only two or therr poins
- It would be interesting to know if Eudragit E-100 alone is able to self aggregate in the different media used for nanocarriers preparation. Did the authors make DLS experiments in these systems?
Author Response
Article: Decrease of Antimicrobial Resistance through Polyelectrolyte-coated Nanoliposomes Loaded with β-Lactam Drug
Response to Reviewer 1
The authors thank the reviewer for the comments and suggestions made. Likewise, we indicate that the suggested changes will be reported in red to facilitate review. At last, the manuscript was submitted to a language editing service (the certificate is attached).
Comments and Suggestions for Authors
This manuscript reports on enhanced efficiency of ampicillin when this antibacterial drug is encapsulated into DOPE nanocarriers. The experimental design is carried out with accuracy and the results obtained again cultures of S. aureus are very promising.
However:
1. The authors do not explain the choice of this phospholipid and indeed it is rather unusual to use DOPE alone to obtain liposomes, since it is well documented that its packing parameter is not suitable for the formation of stable bilayers (i.e. aggregates with zero local curvature). For this reason, DOPE is usually employed in conjunction with DOPC or other bilayer forming lipids. Nevertheless, DOPE is also very sensitive to pH and it is possible that some of the systems prepared in this work actually possess the structure of closed bilayers. In my opinion the physico-chemical characterization carried out by the authors of this manuscript is nor extended enough to talk about liposomes in all the experimental conditions chosen for preparation. Therefore, I would rather call the carriers nano-aggregates or nano-formulations, since the liposome structure is not ascertained with sufficient evidence.
R/ We agree with the comment and we must indicate that our liposomes were not just formed with DOPE, it was used in addition of cholesterol and Epikuron 200™ and this product is a purified, wax-like phosphatidylcholine of soybean origin, produced by column chromatography. Epikuron, consists of phosphatidylcholine and a small number of accompanying phospholipids. Those three components, DOPE, Cholesterol and Epikuron 200™ have shown a synergistic effect to produce the closed bilayer structure which is characteristic for liposomes. Nevertheless, we are currently working on the TEM micrographs for the liposomes structures that we hope to include in a more phisicochemically characterization article.
2. Another week point of this work is that the authors do not discuss in depth the implications of using a positively charged polymer (Eudragit E-100) for the coating. As all cationic molecules, this polymer is likely to have antibacterial effect by itself, but it is not possible to see the extent of the antimicrobial activity exerted by Eudragit E-100 alone, since the corresponding reference tests on S. aureus are not provided.
R/ Regarding this comment, we must say that the experiments about the Eudragit E-100 antibacterial effect were carried out in other experiment, where we did not find any activity, or the minimal inhibition concentration was too high for the evaluated strains. We could provide this information as supplementary material as the reviewer suggest.
Strain | MIC of Eudragit E-100 (Cationic) |
(µg/mL) | |
S. aureus ATCC 25923 | ≥512 |
S. aureus ATCC 29213 | ≥512 |
S. aureus ATCC 43300 | ≥512 |
Also, we have included this evaluation in the figure 5 to give more clarity respect this matter, as the reviewer has suggested.
3. In summary, this work is likely to be of interest for researcher working on pharmaceutical preparations and corresponding antibacterial trials, but the above-mentioned questions have to be addressed before publication.
R/ We agree with the comment and have made the suggested changes. We would like to highlight that the pharmaceutical preparation in which we are projecting these liposomes could be formulated is an oral or dermal formulation. For this reason, we have been working on the evaluation of hemolytic activity for liposomes coated by Eudragit. We could include these results as supplementary material if it is necessary. Moreover, we have included the declaration of this strategy of liposomes coated with polymers is useful projecting pharmaceutical preparations for example for reconstitution pharmaceutical dosage forms (oral or dermal administration) (lines 50-53)
Minor points
4. Though the subject of antibiotic resistance is a major concern in modern pharmacetics, the idea of encapsulationg antibiotics into lipid based nanocarriers is not new and the literature cited by the autors in this respect is rather incomplete.
R/ We believe that we have included enough references to support our work in this respect. We have selected very relevant references including liposomes rather than all kind of nanocarriers and antibiotics rather than all kind of molecules, at the end we got more than 30 references for this article. For instance:
- Drulis-Kawa, Z. & Dorotkiewicz-Jach, A. Liposomes as delivery systems for antibiotics. Int. J. Pharm. 387, 187–198 (2010).
- Pinto-Alphandary, H., Andremont, A. & Couvreur, P. Targeted delivery of antibiotics using liposomes and nanoparticles: Research and applications. Int. J. Antimicrob. Agents 13, 155–168 (2000).
- Kalhapure, R. S., Suleman, N., Mocktar, C., Seedat, N. & Govender, T. Nanoengineered drug delivery systems for enhancing antibiotic therapy. Journal of Pharmaceutical Sciences 104, 872–905 (2015)
- Daraee, H., Etemadi, A., Kouhi, M., Alimirzalu, S. & Akbarzadeh, A. Application of liposomes in medicine and drug delivery. Artif. Cells, Nanomedicine, Biotechnol. 44, 381–391 (2016).
- Nevertheless, we are working on a review type manuscript where we are considering more than 90 references for the assessment of this topic.
5. Figure 1 could be more conveniently replaced by a table, since it doesn't make much sense to show graphs with only two or three points
R/ We thanks for the comment, and we would prefer to keep the figure due to it is important to notice the different behavior of the high and low levels of each one of the study variables.
6. It would be interesting to know if Eudragit E-100 alone is able to self-aggregate in the different media used for nanocarriers preparation. Did the authors make DLS experiments in these systems? https://www.ncbi.nlm.nih.gov/pubmed/28024603
R/ We agree with this comment and it is important to mention that this question was already addressed for our research laboratory in the manuscript entitled: “Physicochemical characterization of in situ drug-polymer nanocomplex formed between zwitterionic drug and ionomeric material in aqueous solution. Mater Sci Eng C Mater Biol Appl. 2017 Mar 1;72:405-414. doi: 10.1016/j.msec.2016.11.097. Epub 2016 Nov 26.”
Abstract:
Biocompatible polymeric materials with the potential to form functional structures, in association with different therapeutic molecules, in physiological media, represent a great potential for biological and pharmaceutical applications. Therefore, here the formation of a nano-complex between a synthetic cationic polymer and model drug (ampicillin trihydrate) was studied. The formed complex was characterized by size and zeta potential measurements, using dynamic light scattering and capillary electrophoresis. Moreover, the chemical and thermodynamically stability of these complexes were studied. The ionomeric material, here referred as EuCl, was obtained by equimolar reaction between Eudragit E and HCl. The structural characterization was carried out by potentiometric titration, FTIR spectroscopy, and DSC. The effect of pH, time, polymer concentration and ampicillin/polymer molar ratio over the hydrodynamic diameter and zeta potential were established. The results show that EuCl ionomer in aqueous media presents two different populations of nanoparticles; one of this tends to form flocculated aggregates in high pH and concentrations, by acquiring different conformations in solution by changing from a compact to an extended conformation. Moreover, the formation of an in situ interfacial polymer-drug complex was demonstrated, this could slightly reduce the hydrolytic degradation of the drug while affecting its solubility, mainly under acidic conditions
Reviewer 2 Report
MDPI pharmaceuticals: Decrease of Antimicrobial resistance ..
The manuscript addresses the preparation of polyelectrolyte-coated nanoliposomes loaded with ampicillin, and their activity against reference strains of Staphylococcus aureus. The coating process did not significantly affect the encapsulation of ampicillin. The stability of liposomes, defined as resistance to adsorption and auto-aggregation, was increased by coating with polymer.
Antimicrobial susceptibility was performed with broth microdilution according to CSLI guidelines. Three ATCC reference strains without and with different resistance mechanisms were used. Encapsulation of ampicillin in liposomes marginally lowered the MICs of all strains, while coating with polymer was considerably more effectual. Interestingly, the effect was most prominent on ATCC_43300, a strain that possesses both the capability of degrading ampicillin (blaZ), and an altered antimicrobial target (PBP2a). The difference between strains with single-and dual-resistance mechanisms is not readily explained and most be pursued in further analysis.
The study is interesting and may have bearing for optimal administration of antimicrobials when treating infections caused by multi-resistant organisms.
Spécific points.
The abbreviations AMP, AMP-NCL and AMP-CL are described in Figure legends (Figs 3 and 5), but should be introduced when first mentioned in the text (Line 150).
Author Response
Article: Decrease of Antimicrobial Resistance through Polyelectrolyte-coated Nanoliposomes Loaded with β-Lactam Drug
Response to Reviewer 2
The authors thank the reviewer for the comments and suggestions made. Likewise, we indicate that the suggested changes will be reported in red to facilitate review. At last, the manuscript was submitted to a language editing service (the certificate is attached).
Comments and Suggestions for Authors
The manuscript addresses the preparation of polyelectrolyte-coated nanoliposomes loaded with ampicillin, and their activity against reference strains of Staphylococcus aureus. The coating process did not significantly affect the encapsulation of ampicillin. The stability of liposomes, defined as resistance to adsorption and auto-aggregation, was increased by coating with polymer.
Antimicrobial susceptibility was performed with broth microdilution according to CSLI guidelines. Three ATCC reference strains without and with different resistance mechanisms were used. Encapsulation of ampicillin in liposomes marginally lowered the MICs of all strains, while coating with polymer was considerably more effectual. Interestingly, the effect was most prominent on ATCC_43300, a strain that possesses both the capability of degrading ampicillin (blaZ), and an altered antimicrobial target (PBP2a). The difference between strains with single-and dual-resistance mechanisms is not readily explained and most be pursued in further analysis.
The study is interesting and may have bearing for optimal administration of antimicrobials when treating infections caused by multi-resistant organisms.
Specific points.
1. The abbreviations AMP, AMP-NCL and AMP-CL are described in Figure legends (Figs 3 and 5), but should be introduced when first mentioned in the text (Line 150).
R/ We agree with the comment and we have done the respectively correction
Round 2
Reviewer 1 Report
I found the the authors addressed the main questions raised and the revised manuscript is acceptable for publication in Pharmaceuticals. I only recommend to add in text that (cryo?) TEM investigation is in progress to improve the physico-chemical characterization of these nanocarriers and assess their liposomal structure